# High Burden of Hepatitis B Virus and Occult Infection Among HIV-Positive Adults and Pregnant Women in Southwest Cameroon

**DOI:** 10.3390/pathogens14111128

**Published:** 2025-11-05

**Authors:** Macqueen Ngum Mbencho, Le Chi Cao, Eric A. Achidi, Stephen Mbigha Ghogomu, Thirumalaisamy P. Velavan

**Affiliations:** 1Institute of Tropical Medicine, German Center for Infection Research (DZIF), University of Tübingen, 72074 Tübingen, Germany; mcqueengum@gmail.com (M.N.M.); lechicao@hueuni.edu.vn (L.C.C.); 2Molecular and Cell Biology Laboratory, University of Buea, Buea P.O. Box 63, Cameroon; stephen.ghogomu@ubuea.cm; 3Hue University of Medicine and Pharmacy, Hue University, Hue City 530000, Vietnam; 4Faculty of Sciences, University of Buea, Buea P.O. Box 63, Cameroon; achidi_e@yahoo.com; 5Vietnamese German Center for Medical Research (VG-CARE), Hanoi 100000, Vietnam; 6Faculty of Medicine, Duy Tan University, Da Nang 550000, Vietnam

**Keywords:** chronic hepatitis B, occult hepatitis, pregnant women, HIV, Cameroon

## Abstract

Chronic hepatitis B virus (HBV) and Occult HBV infection (OBI) remain a health burden in sub-Saharan Africa. This study investigated HBV prevalence, circulating genotypes, and associated risk factors with HBV exposure among HIV-positive adults on antiretroviral therapy and pregnant women in southwestern Cameroon. A total of 233 HIV patients and 190 third-trimester pregnant women were screened for HBV DNA, viral load, serological markers (HBsAg, anti-HBc, and anti-HBs), and HBV genotypes were determined by partial sequencing of the S gene. HBV DNA was detected in 10% of HIV-positive patients and 4% of pregnant women, with an overall prevalence of 7%. OBI accounted for 9% and 3%, respectively. Anti-HBc seroprevalence was high (75% in HIV, 46% in pregnant women), while self-reported vaccination coverage was low (1% and 11%). Genotypes A, B, D, and E were identified, with genotype B reported for the first time in Cameroon. Immune escape mutations and the adefovir resistance mutation rtA181V were detected. Self-reported alcohol use was associated with HBV exposure in HIV patients (aOR = 2.08; *p* = 0.028) and inversely associated with tertiary education in pregnant women (aOR = 0.18; *p* = 0.038). This study highlights a significant burden of HBV and OBI among vulnerable populations in Cameroon.

## 1. Introduction

Chronic hepatitis B virus (HBV) infection is a global health concern, disproportionately affecting individuals living in sub-Saharan Africa and the Western Pacific regions [1]. Of the estimated 254 million people living with chronic HBV worldwide, over 64 million reside in Africa, including approximately 3.6 million children under five years of age [2]. Between 2019 and 2022, global HBV-related deaths rose from 820,000 to 1.1 million, primarily due to cirrhosis and hepatocellular carcinoma (HCC) [1]. Despite the availability of an effective HBV vaccine for over four decades, mother-to-child transmission continues to sustain HBV endemicity in sub-Saharan Africa [3]. Nearly one-quarter of HBV-infected individuals in Africa are under 20 years of age [2]. Without timely intervention, 80–90% of neonates infected perinatally and 30–50% of children infected between one and five years of age progress to chronic infection compared to 5–10% in those infected as adults [4]. In 2022, only 5% of individuals with chronic HBV in Africa were diagnosed, and a mere 2% were receiving treatment [1]. Among pregnant women, treatment coverage was below 1%, while timely birth-dose vaccine coverage stood at only 14% [2].

The human immunodeficiency virus (HIV) and HBV co-infection presents an additional challenge, with sub-Saharan Africa accounting for approximately 1.9 million of the 2.7 million people co-infected [5]. The overlap in transmission routes and limited screening strategies contributes to the persistence of co-infection. Moreover, HIV infection accelerates HBV-related liver disease progression, increasing the risk of fibrosis and HCC, even among those on antiretroviral therapy (ART) [6]. Cameroon has a national HBV seroprevalence of approximately 11% for hepatitis B surface antigen (HBsAg) [7], with high rates among HIV-positive individuals (12%) and pregnant women (≥6%) [8,9]. Although infant HBV vaccination has been part of the national immunization schedule since 2005, the country lacks a coordinated strategy for adult HBV screening, treatment, and long-term care [10]. The World Health Organization (WHO) guidelines emphasize the importance of targeted screening in high-risk populations such as people living with HIV and pregnant women, who are at greater risk of end-stage liver disease when co-infected.

While HBsAg testing is commonly used for HBV screening, it may fail to detect cases of occult hepatitis B infection (OBI), a condition in which HBV DNA is present at very low levels (<200 IU/mL) in blood despite a negative HBsAg result [11]. In OBI, the episomal covalently closed circular DNA (cccDNA) within infected hepatocytes remains transcriptionally active at a low rate, resulting in reduced HBsAg expression and secretion, often falling below the detection limit of most commercial HBsAg assays with low analytical sensitivity (approximately 0.05 IU/mL). In addition, mutations within the surface gene (S gene) can alter antigenic sites, leading to diagnostic escape [11]. In our previous study among Cameroonian blood donors, we reported an OBI prevalence of 5%, underscoring the magnitude of undetected HBV transmission [12]. This study aimed to assess the prevalence of HBV infection, OBI, and distribution of genotypes, and associated risk factors among two vulnerable populations, including people living with HIV receiving ART and third-trimester pregnant women, in the Southwest region of Cameroon. Our findings aim to inform national policy on HBV control and advocate for expanded diagnostics and vaccination strategies in high-risk groups.

## 2. Materials and Methods

### 2.1. Study Design and Ethical Approval

A cross-sectional study was conducted from 21 March to 30 June 2023, in the Southwest region of Cameroon. The study was approved by the Institutional Review Board of the University of Buea, Cameroon (Reference: 2022/1849-10/UB/SG/IRB/FHS) and by the Ethics Committee of the University of Tübingen, Germany (Reference: 379/2023B02). Administrative clearance was obtained from the Southwest Regional Delegation of Public Health. Written informed consent was obtained from all participants before enrolment.

### 2.2. Study Population and Sampling

In this study, 233 HIV-positive adults undergoing ART were recruited consecutively after obtaining their consent during their routine follow-up appointments at the HIV treatment departments of the Regional Hospitals in Buea and Limbe. No records of liver disease were available for these patients. Also, 190 pregnant women in their third trimester who were receiving antenatal care at integrated health centers in Buea (PMI) and Mutengene (CMA) were consecutively enrolled in the study after providing written consent. None of the HIV patients were pregnant, and none of the pregnant women had a record of HIV infection. For each participant, 3 mL of venous blood was collected, and serum was separated, preserved in DNA/RNA Shield (Zymo Research, Irvine, CA, USA), and stored at −20 °C for subsequent molecular and serological analyses. Socio-demographic and clinical information were obtained using structured questionnaires.

### 2.3. HBV DNA Detection and Quantification

Viral DNA was extracted from 200 μL of serum using the QIAamp Viral DNA Mini Kit (Qiagen, Hilden, Germany) according to the manufacturer’s instructions. Nested polymerase chain reaction (PCR) targeting the conserved overlapping S/P region of the HBV genome was used to qualitatively detect HBV DNA, as previously described [12] (Appendix A). PCR-positive samples were subsequently analyzed by real-time PCR using a TaqMan probe-based assay targeting a conserved 90 bp segment within the S gene (nucleotide positions 182–271, GenBank accession: X75657), as previously described [12] (Appendix A). Quantification was performed using the QuantiTect Probe PCR Master Mix (Qiagen) on a LightCycler 480 II (Roche, Mannheim, Germany). The assay detection limit was 12 IU/mL, established using tenfold serial dilutions of a control plasmid (10^6^ copies/μL). HBV viral load was calculated from cycle threshold (Ct) values based on a standard curve.

### 2.4. Serological Testing

All participants were screened for antibodies against the HBV core antigen (total anti-HBc) using the qualitative Monolisa™ anti-HBc PLUS assay (Bio-Rad, Hercules, CA, USA). HBV DNA-positive individuals were additionally tested for HBsAg using the Determine™ HBsAg 2 rapid test (Abbott Laboratories, Abbott Park, IL, USA), and for antibodies against the surface antigen (anti-HBs) using the Monolisa™ anti-HBs PLUS assay (Bio-Rad). Anti-HBs titers > 10 mIU/mL were considered positive. Absorbance readings were taken using a CLARIOstar microplate reader (BMG Labtech, Ortenberg, Germany). All assays were performed following the manufacturer’s protocols.

### 2.5. HBV Genotyping and Phylogenetic Analysis

Amplicons from HBV DNA-positive samples were purified using the Exo-SAP-IT kit (USB, Affymetrix, Santa Clara, CA, USA) and sequenced using the BigDye™ Terminator v3.1 Cycle Sequencing Kit (Applied Biosystems, Foster City, CA, USA) on an ABI 3130xl Genetic Analyzer (Applied Biosystems, Foster City, CA, USA). Consensus sequences were assembled using SeqMan (DNASTAR Lasergene). A reference set of 47 HBV sequences representing genotypes A–I was included. Multiple sequence alignment was performed using MAFFT v7 (G-INS-i algorithm) [13], and the phylogenetic tree was reconstructed using MEGA v12 [14] with the Neighbor-joining method and the Kimura 2-parameter model with gamma distribution (K2+G). Bootstrapping (1000 replicates) assessed the robustness of phylogenetic assembly. The phylogenetic tree was annotated and visualized using the Interactive Tree of Life (iTOL) version 6 [15].

Mutation analysis of the surface (S) gene and polymerase (P) gene was performed using BioEdit v7 and the Geno2pheno [hbv] online tool (https://hbv.geno2pheno.org/) (accessed on 1 May 2025). Sequences were compared to genotype-specific references from GenBank (Genotype A: M57663, AF090842; Genotype B: AB642091; Genotype D: M32138; Genotype E: X75657, AB032431). Nonsynonymous mutations were annotated, with a focus on immune escape variants and drug resistance markers. Additionally, the mutations observed were categorized according to whether they occurred in HBV or OBI cases. Sequences generated in this study were submitted to GenBank under accession numbers PV604144–PV604172.

### 2.6. Statistical Analysis

Statistical analyses were performed using R version 4.0. Continuous variables were summarized using medians and interquartile ranges (IQR), while categorical variables were expressed as frequencies and percentages. The Pearson chi-square test and Fisher’s exact test were initially used to assess associations between categorical variables, with the latter applied when expected cell counts were ≤5. Logistic regression models were then applied to preselected variables with a *p*-value threshold of ≤0.2 to evaluate factors associated with HBV exposure (anti-HBc positivity). Adjusted odds ratios (aOR) with 95% confidence intervals (CI) were calculated. A *p*-value < 0.05 was considered statistically significant

## 3. Results

### 3.1. Participant Demographics

A total of 423 individuals were enrolled, comprising 233 HIV-positive adults and 190 third-trimester pregnant women. Among the HIV cohort, 83% were female, with a median age of 46 years (interquartile range [IQR]: 39–55). In the pregnant women cohort, the median age was 27 years (IQR: 24–31). HBV vaccination coverage was low: 1% in HIV patients and 11% in pregnant women (Table 1).

### 3.2. Prevalence of HBV Infection

HBV DNA was detected in 10% (24/233) of HIV patients and 4% (7/190) of pregnant women, corresponding to an overall prevalence of 7.3% (31/423) (Table 1). Occult hepatitis B infection, defined as HBV DNA positivity with negative HBsAg, was identified in 9% (21/233) of HIV patients and 3% (5/190) of pregnant women (Table 1). HBsAg positivity was 1% (HIV: 3/233; pregnant women 2/190) in both cohorts (Table 1).

### 3.3. HBV Serological Markers and Viraemia

The seroprevalence of anti-HBc, a marker of past or ongoing HBV exposure, was 75% (174/233) in HIV patients and 46% (87/190) in pregnant women, yielding an overall exposure rate of 62% (261/423) (Table 1). In OBI-positive HIV patients, 86% (18/21) were anti-HBc-positive, of whom 29% (6/21) were also anti-HBs-positive. Among OBI-positive pregnant women, 20% (1/5) were anti-HBc-positive, and none of them had detectable anti-HBs antibodies. Isolated anti-HBc seropositivity (anti-HBc^+^/anti-HBs^−^) was found in 57% of OBI-positive HIV patients and in 20% of OBI-positive pregnant women. In individuals with detectable anti-HBs, the mean anti-HBs titer was 167.7 mIU/mL (standard deviation: 133.9) and ranged from 20 to 384 mIU/mL (Table 1). All five HBsAg-positive individuals (three HIV patients and two pregnant women) had detectable HBV DNA levels ranging from 2.46 × 10^2^ to 7.67 × 10^4^ IU/mL, whereas all OBI cases (n = 26) in both cohorts had HBV DNA levels that were either undetectable or at the assay’s lower limit of detection (Table 1). In the study population, 1% of HIV patients and 11% of pregnant women reported having received HBV vaccination.

### 3.4. HBV Genotypes and Mutations

Phylogenetic analysis of the sequenced S gene fragments identified HBV genotypes A, B, D, and E. Among HIV patients, genotype A was predominant (14/24), followed by genotype B (7/24), with one genotype E infection (Figure 1). Genotype B was detected exclusively in OBI-positive cases and represents the first report of this genotype in Cameroon. Two HIV samples could not be genotyped due to sequencing failure. In pregnant women, genotype E was most common (5/7), followed by one case each of genotypes A and D. Of the seven genotyped samples, four genotype E infections were OBI-positive (Figure 1).

Mutation analysis revealed several nonsynonymous substitutions in the major hydrophilic region (MHR) of the S gene, including sP120T and sT126A, both associated with HBsAg immune escape and OBI. The sF200Y mutation was observed in all genotype B OBI cases. In the reverse transcriptase (RT) region, the rtA181V mutation associated with adefovir resistance was found in one HIV patient. Additional non-synonymous substitutions (sY200F, rtW153R) were identified in both HBsAg-positive and negative genotype A sequences (Table 2).

### 3.5. Factors Associated with HBV Exposure

In HIV patients, self-reported alcohol consumption and HIV viral load were significantly associated with anti-HBc positivity (Table 3). Alcohol consumers had a higher prevalence of HBV exposure (80%) compared to non-consumers (64%) (*p* = 0.0115). Additionally, anti-HBc positivity was more frequent in participants with undetectable HIV viral loads (79%) than those with low (53%) or high (75%) viral loads (*p* = 0.0008) (Table 3). Also, there was no association between OBI status and HIV viral load. Among pregnant women, marital status, education level, and parity were significantly associated with HBV exposure (Table 3). Married women had a higher anti-HBc positivity rate (56%) than single women (33%) (*p* = 0.0013) (Table 3). Women with only primary education showed higher exposure (82%) than those with tertiary education (38%) (*p* = 0.0315) (Table 3). Multigravida women had higher exposure (57%) compared to primigravida (38%) (*p* = 0.0139) (Table 3). Multivariate logistic regression confirmed that alcohol use was independently associated with HBV exposure in HIV patients (adjusted odds ratio [aOR] = 2.08; 95% CI: 1.08–4.00; *p* = 0.028) (Table 4). In pregnant women, tertiary education was protective (aOR = 0.18; 95% CI: 0.03–0.78; *p* = 0.038), and marital status remained borderline significant (aOR = 2.00; 95% CI: 1.01–4.00; *p* = 0.048) (Table 4).

## 4. Discussion

This study provides updated insights into the prevalence, genotype distribution, and risk factors associated with HBV infection among two high-risk populations: HIV-positive adults and pregnant women in Southwestern Cameroon. Despite national vaccination efforts, the findings reveal persistent gaps in HBV prevention and underscore the importance of enhanced screening and targeted public health interventions.

We report an overall HBV DNA prevalence of 7%, with a notably higher rate among HIV patients (10%) compared to pregnant women (4%). These figures are consistent with regional and global estimates for HIV/HBV co-infection and pregnant populations, particularly in sub-Saharan Africa [16]. Notably, the prevalence of hepatitis B surface antigen (HBsAg) was only 1% in both cohorts, emphasizing the limitations of HBsAg-based screening alone. A significant proportion of HBV infections identified in this study were occult hepatitis B infections defined as HBV DNA positivity despite negative HBsAg. OBI was observed in 9% of HIV patients and 3% of pregnant women. These rates are higher than our previously reported 5% OBI prevalence in blood donors from Cameroon [12] and are within the expected range for people living with HIV in Africa (8–26%) [17]. The high OBI burden in HIV patients may be influenced by immunosuppression and antiretroviral therapy, which can suppress HBsAg expression without clearing HBV DNA.

The seroprevalence of anti-HBc, a marker of past or current HBV exposure, was strikingly high: 75% among HIV patients and 46% in pregnant women in this study, suggesting substantial transmission within the population. While anti-HBs levels were not measured for the entire study population, the high anti-HBc prevalence may suggest low vaccine-derived immunity among adults. Indeed, from self-reported vaccination history, only about 1% of HIV patients and 11% of pregnant women reported having received HBV vaccination despite their high-risk status. In Cameroon, vaccination coverage for at least one dose reaches 99% among individuals born after the vaccine’s inclusion in the national infant immunization program in 2005 [18]. However, coverage among the general adult population remains around 5% [19]. These findings highlight the lack of comprehensive vaccination strategies for adults in Cameroon and the on-going need to strengthen implementation of the WHO-recommended birth dose to prevent early-life infection and subsequent transmission.

HBV genotypes A, B, D, and E were detected in the study population. Genotype A predominated among HIV patients, whereas genotype E was more common among pregnant women, with both representing the dominant genotypes in the region. Notably, genotype B was identified exclusively in HIV patients, and all were OBI-positive, constituting the first documented detection of this genotype in Cameroon. The emergence of genotype B may be related to cross-border migration, travel, or demographic shifts. Mutations in the S gene, including sP120T and sT126A, were identified in multiple OBI cases. These mutations, located in the major hydrophilic region (MHR) of HBsAg, are associated with immune escape and diagnostic failure [20,21]. The sF200Y mutation was observed in all genotype B sequences. In the polymerase gene, we detected rtA181V, a known drug resistance mutation associated with reduced susceptibility to adefovir [22] and potential cross-resistance to tenofovir [23]. Its presence in an HIV-positive patient receiving Tenofovir Disoproxil Fumarate/Lamivudine/Dolutegravir (TDF/3TC/DTG) raises concern for antiviral resistance and calls for closer monitoring.

In the HIV cohort, self-reported alcohol use was significantly associated with HBV exposure (anti-HBc positivity), after adjusting for confounders. Alcohol use may reflect underlying behavioral risk factors, such as unprotected sex or injection drug use that expose individuals to HBV, rather than a direct virological effect [24]. However, the type, amount, or frequency of alcohol was not assessed, limiting the ability to examine a potential dose–response relationship. Similarly, in pregnant women, lower education, marital status, and multiparity were associated with higher HBV exposure. Tertiary education appeared protective, potentially due to greater awareness and health-seeking behavior.

Our findings underscore several urgent public health priorities for HBV control in Cameroon and similar settings. Routine screening must extend beyond HBsAg testing to include anti-HBc and nucleic acid testing, particularly among high-risk groups such as pregnant women and HIV patients, to ensure early detection and appropriate clinical management. Integrating nucleic acid testing into existing HIV programs would also support optimal ART regimen selection and help avoid drug combinations that are unsuitable for HBV co-infected individuals. Furthermore, it appears OBI may persist independently of HIV replication control, emphasizing the importance of continued HBV monitoring regardless of viral suppression status to prevent HBV reactivation. Expanding vaccination coverage is also critical, with emphasis on administering the birth dose promptly, implementing catch-up immunization for adolescents and adults, and prioritizing vaccination for healthcare workers, HIV-positive individuals, and expectant mothers. Continuous surveillance of circulating HBV genotypes and resistance-associated mutations is necessary to guide diagnostic strategies, assess vaccine effectiveness, and inform treatment decisions. Furthermore, health education initiatives that address underlying behavioral risk factors, including alcohol consumption and limited awareness, can significantly reduce transmission and enhance the effectiveness of prevention programs.

Several limitations must be considered. First, the cross-sectional study design limits the causal interpretation of the observed associations. Therefore, the relationships between HBV markers and factors such as alcohol use or educational level should be considered correlational rather than causal. Second, anti-HBs were only measured in HBV DNA-positive individuals, limiting assessment of population-level immunity. As such, the self-reported vaccination rates may not reflect actual protective immunity. Third, clinical parameters such as liver function tests or fibrosis scores were not included, restricting the interpretation of the clinical significance of HBV infections. Finally, while phylogenetic analysis identified genotypes and mutations, whole-genome sequencing could provide a more comprehensive picture of viral diversity.

## 5. Conclusions

This study highlights a considerable burden of HBV, including a high rate of occult infection and the first documentation of genotype B in Cameroon. Strengthened policies on HBV screening, adult vaccination, and education, particularly in high-risk groups, are urgently needed to meet global hepatitis elimination goals.

## Figures and Tables

**Figure 1 pathogens-14-01128-f001:**
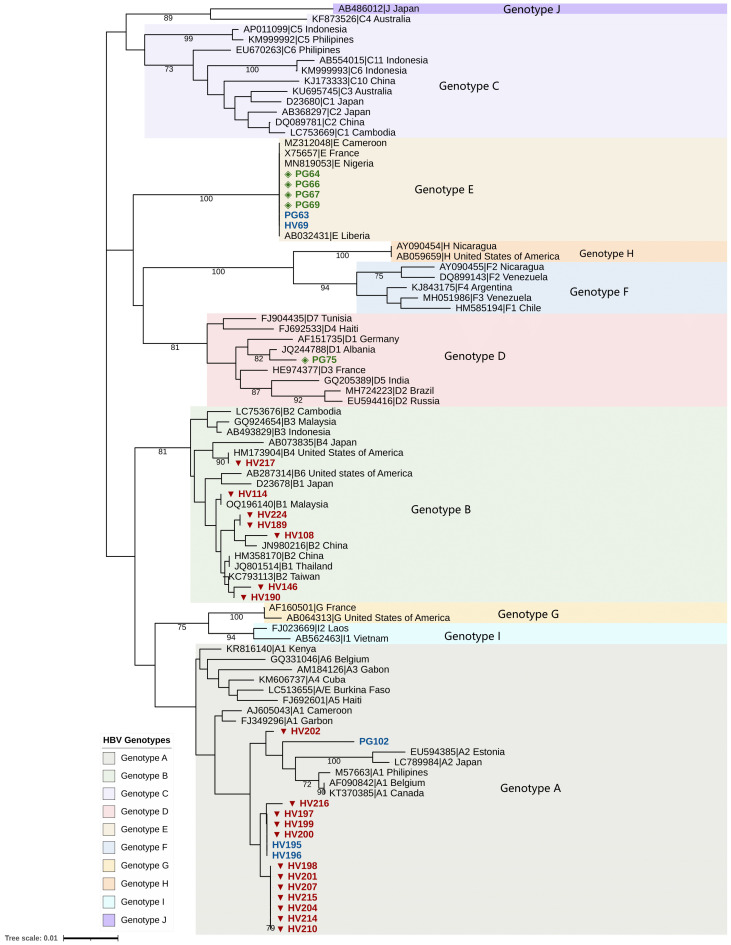
Phylogenetic tree of the S/P region of HBV sequences from study participants. Sample identifiers are labeled as HV (person with HIV) and PG (pregnant woman). Occult HBV infection sequences are indicated by red triangles (persons with HIV, n = 19) and green diamonds (pregnant women, n = 5). HBsAg-positive sequences are indicated in blue (HIV; n = 3; Pregnant women; n = 2). Sequences from individuals living with HIV clustered with HBV genotypes A and B, while sequences from pregnant women clustered with genotypes A, D, and E. A total of 47 reference genotypes A–I were included in the analysis. Bootstrap values are shown at the nodes and represent percentages from 1000 replicates; only values greater than 70% are displayed. Of the 332 bp fragment of the S/P region amplified, a sequencing coverage of 328 bp was achieved for the successfully sequenced samples. Two of the 24 samples from HIV-positive participants could not be genotyped due to sequencing failure, whereas all samples from pregnant women (n = 7) were successfully genotyped.

**Table 1 pathogens-14-01128-t001:** Overall prevalence of anti-HBs and anti-HBc markers and combination.

Characteristics	Persons with HIV (n = 233) (%)	Pregnant Women (n = 190) (%)
Gender (Female)	194 (83)	190 (100)
Median Age (IQR)	46 (39–55)	27 (24–31)
Age ≤ 20	1 (1)	17 (9)
Age 21–40	67 (29)	169 (89)
Age 41–60	134 (58)	4 (2)
Age > 60	31 (13)	0
anti-HBc-positive	174 (75)	87 (46)
HBV DNA Positive	24 (10)	7 (4)
^†^ HBV viral load (IU/mL) median [Range]	12.4 (12.4–76,700)	26,523 (246–52,800)
OBI Positive (HBV DNA+ve, HBsAg-ve)	21 (9)	5 (3)
HBsAg Positive	3 (1)	2 (1)
^‡^ anti-HBs-positive mIU/mL [Range]	6 (25) (20–384)	0 (0)
HIV viral load range (copies/mL)	Undetectable to 401,538	Not available
HBV genotypes	A n = 14 (58) B n = 7 (29) E n = 1 (4)	A n = 1 (14) D n = 1 (14) E n = 5 (71)

^†^ detectable viral load (HIV: n = 8/24; Pregnant women: n = 2/7), ^‡^ anti-HBs-positive measured only in HBV DNA positives (HIV: n = 6/24; Pregnant women: n = 0/7); IQR: Inter Quartile Range, HBV: Hepatitis B virus; OBI: Occult hepatitis B infection; HBsAg: Hepatitis B Surface Antigen; anti-HBc: hepatitis B core antibody.

**Table 2 pathogens-14-01128-t002:** Mutations in HBV Surface (S) Gene and Polymerase (P) RT Domain.

Study Cohort	HBV Genome	Amino Acid Substitution	HBV Genotypes
HIV patients (n = 233)	‘S’ gene that encodes the Hepatitis B surface antigen (HBsAg)	sY200F	A
		sS207T *	
		sF200Y *	B
		sT126A *	
		sP120T *	
		sK122R *	
	‘P’ gene that encodes the viral DNA polymerase, essential for the replication of the viral genome.	rtW153R	A
		rtQ215H	
		rtD131N *	B
		rtN134S *	
		rtN134D *	
		rtA181V *	
		rtI187V *	
Pregnant Women (n = 190)	‘S’ gene that encodes the Hepatitis B surface antigen (HBsAg)	sK122R	A
		sP135A	
		sG130S *	D
		sL193S *	

* Mutations found in OBI cases. No mutations were observed in genotype E in both study cohorts.

**Table 3 pathogens-14-01128-t003:** Association between demographic/clinical factors and anti-HBc positivity (HIV and pregnant women cohorts).

Study Cohort	Demographic and Clinical Factors *	Anti-HBc Positive (%)	*p*-Value
HIV cohort (n = 233)	Alcohol consumption	80%	0.0115
	HIV viral load (undetectable)	79%	0.0008
Pregnant women (n = 190)	Marital status	56% (married)	0.0013
	Education level	82% (primary)	0.0315
	Parity (≥2 births)	57%	0.0139

* Only variables with significant associations are shown.

**Table 4 pathogens-14-01128-t004:** Multivariate logistic regression for factors associated with HBV exposure.

Variables *	Adjusted Odds Ratio (aOR) Confidence Interval (95% CI)	*p*-Value
Alcohol use (among persons with HIV)	2.08 (1.08–4.00)	0.028
Married (pregnant women)	2.00 (1.01–4.00)	0.048
Tertiary education (pregnant women)	0.18 (0.03–0.78)	0.038

* Only variables independently associated with HBV exposure are shown.

## Data Availability

All data generated in this study are included in this article. All obtained sequences were available in the NCBI GenBank database (https://www.ncbi.nlm.nih.gov/genbank/) (accessed on 5 May 2025) with the following accession numbers: PV604144 to PV604172.

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
