# Peer review of "High Burden of Hepatitis B Virus and Occult Infection Among HIV-Positive Adults and Pregnant Women in Southwest Cameroon"

_pathogens, 2025, doi:10.3390/pathogens14111128_

Round 1
Reviewer 1 Report
Comments and Suggestions for Authors
This is a strong and timely epidemiological and molecular study addressing a critical intersection of HBV, HIV, and maternal health in sub-Saharan Africa.
major points:
- Clinical interpretation of OBI: While the detection of occult infections is important, the absence of accompanying liver-function or HIV-viral-load data limits interpretation of clinical impact. Including ALT/AST values or referencing the potential implications for ART co-management would strengthen translational relevance.
- Cross-sectional design constraints: The study design precludes causal inference regarding ART exposure or behavioral risk factors. A brief acknowledgment of this limitation in the discussion would clarify that associations (e.g., alcohol use, education) are correlational, not causal.
- Vaccine coverage interpretation: The authors rightly note low vaccination rates but rely on self-reported data. Clarifying whether anti-HBs titers were used to confirm vaccination, or distinguishing between natural and vaccine-induced immunity, would improve reliability of this conclusion.
- Public-health framing:
The discussion could more explicitly link findings to feasible interventions, such as integrating maternal HBV screening into antenatal care, or combining HBV testing with existing HIV programs to identify patients that should not use for example dolutegravir/3TC to strengthen the policy relevance of the manuscript.
Minor points:
1) The abstract could briefly mention the first report of genotype B to emphasize novelty.
2) Clarify whether “occult infection” refers strictly to HBsAg-negative, HBV DNA-positive cases (<200 IU/mL).
3) Figure legends could specify sequencing coverage and the number of samples successfully genotyped.
Reviewer 2 Report
Comments and Suggestions for Authors
The study by Macqueen Ngum Mbencho et al. is of great interest, but the article's wording is very confusing and requires extensive re-reading to understand the relevance of the results presented.
It is not clear why anti-HBs was only measured in HBV DNA-positive individuals. This represents a significant limitation, as the authors indicate, but it must be resolved by retesting the entire study population for anti-HBs.
Regarding OBI, which as the authors indicate can be defined as HBV DNA is present despite negative HBsAg status and is characterized by low-level viremia (typically <200 IU/mL), the authors indicate that "can be missed by conventional serological tests." This statement is not understood. To which serological test is this referring? Please clarify whether this refers to the fact that HBsAg may not be detected due to possible sensitivity issues.
In the introduction, it is indicated that Cameroon has a national HBV seroprevalence of approximately 11%. Are they referring to antiHBc positive or HBsg positive? It is very important to distinguish between the two. The presence of infection, that is, HBsAg positivity, and the seroprevalence of only anti-HBc should be made clear. It is also important to indicate the proportion of patients theoretically immunized by the presence of antiHBs.
-in 2.2 Serological assays. No It does not detail at all what these assay are, which are detailed in section 2.4. This is very confusing. Section 2.4 should be included in 2.2.
The patient selection criteria are not clearly indicated. In section 2.2, serological assessments, it indicates that they are HIV-positive on antiretroviral treatment. Are they randomized, or from a hepatitis clinic, etc.? And are pregnant women HIV-negative? They should be included in a specific patient section in section 2.1. This is extremely important because the percentage of anti-HBc positive patients is extremely high (75%). As mentioned, it does not indicate whether the women are HIV-positive or on antiretroviral therapy. This can be inferred, but it is not a matter for the reader to deduce. The manuscript should be clear, and in this sense, it is extremely inadequate.
Author Response
Dear Reviewer,
We would like to express our sincere gratitude for your constructive and insightful feedback, which has greatly contributed to improving the quality and clarity of our manuscript. All comments and suggestions have been carefully addressed in the attached document.
Sincerely,
Prof. Thirumalaisamy P. Velavan, on behalf of all co-authors

Round 2
Reviewer 2 Report
Comments and Suggestions for Authors
The manuscript has improved considerably with the modifications in this new version and I believe it is now in a condition to be accepted